# Lead-Free Perovskite Thin Films for Gas Sensing through Surface Acoustic Wave Device Detection

**DOI:** 10.3390/nano14010039

**Published:** 2023-12-22

**Authors:** Nicoleta Enea, Valentin Ion, Cristian Viespe, Izabela Constantinoiu, Anca Bonciu, Maria Luiza Stîngescu, Ruxandra Bîrjega, Nicu Doinel Scarisoreanu

**Affiliations:** 1National Institute for Laser, Plasma and Radiation Physics, 077125 Magurele, Romania or nicoleta.enea@unifi.it (N.E.); cristian.viespe@inflpr.ro (C.V.); izabela.constantinoiu@inflpr.ro (I.C.); anca.bonciu@inflpr.ro (A.B.); maria.stingescu@inflpr.ro (M.L.S.); ruxandra.birjega@inflpr.ro (R.B.); nicu.scarisoreanu@inflpr.ro (N.D.S.); 2Department of Physics and Astronomy, University of Florence, Via G. Sansone 1, 50019 Sesto Fiorentino, FI, Italy; 3Faculty of Physics, University of Bucharest, 077125 Magurele, Romania

**Keywords:** thin film, PLD, MAPLE, BCTZ, lead-free piezoceramic, PEI, CO_2_ and O_2_ detection, SAW sensor

## Abstract

Thin film technology shows great promise in fabricating electronic devices such as gas sensors. Here, we report the fabrication of surface acoustic wave (SAW) sensors based on thin films of (1 − x) Ba(Ti_0.8_Zr_0.2_)O_3−x_(Ba_0.7_Ca_0.3_)TiO_3_ (BCTZ50, x = 50) and Polyethylenimine (PEI). The layers were deposited by two laser-based techniques, namely pulsed laser deposition (PLD) for the lead-free material and matrix assisted pulsed laser evaporation (MAPLE) for the sensitive polymer. In order to assay the impact of the thickness, the number of laser pulses was varied, leading to thicknesses between 50 and 350 nm. The influence of BCTZ film’s crystallographic features on the characteristics and performance of the SAW device was studied by employing substrates with different crystal structures, more precisely cubic Strontium Titanate (SrTiO_3_) and orthorhombic Gadolinium Scandium Oxide (GdScO_3_). The SAW sensors were further integrated into a testing system to evaluate the response of the BCTZ thin films with PEI, and then subjected to tests for N_2_, CO_2_ and O_2_ gases. The influence of the MAPLE’s deposited PEI layer on the overall performance was demonstrated. For the SAW sensors based on BCTZ/GdScO_3_ thin films with a PEI polymer, a maximum frequency shift of 39.5 kHz has been obtained for CO_2_; eight times higher compared to the sensor without the polymeric layer.

## 1. Introduction

In an era where the demand for air quality monitoring to ensure the safety of the environment and human health has increased, gas sensors have become crucial for the detection of various gases (such as volatile organic compounds [1], inorganic gases [2,3,4,5,6,7,8,9,10], etc.). In the last few decades, surface acoustic wave (SAW) technology has been integrated into gas detection devices, providing them with attractive advantages such as compactness, portability, low-cost manufacturing, high sensitivity and accuracy, real-time measurements and good reliability [11,12,13]. Moreover, through a suitable choice of materials for fabricating these types of sensors, they also offer outstanding characteristics like selectivity, reversibility and linearity [14]. An SAW gas sensor (Rayleigh-wave, delay-line type sensor) consists of an expensive piezoelectric single-crystal material (usually ST-Quartz or LiNbO_3_) with interdigital transducer electrodes (IDTs) deposited on its surface and a recognition layer in-between.

SAW devices convert the input electrical signal sent through IDT into a mechanical one, which is subsequently altered by the sensing event and converted back into an electrical one through the output IDT. The changes produced by the exposure of the detection layer to the target analyte consist of variations in the amplitude, frequency and phase of the output signal relative to the input signal [15,16,17].

Regarding the sensing layers, a wide range of materials have been exploited for these SAW devices [11,18,19]. Common materials such as conductive/non-conductive polymers and metal oxides have been chosen most often to recognise gas molecules [20,21]. But other materials have also been tested, including carbon-based nanostructures [22,23,24,25], composite materials [26,27], metals [28], metal-organic frameworks [29] and porous materials [30]. There are some reports in the literature on using a ferroelectric lead zirconium titanate (PZT) thin film as the sensing layer to improve the coupling coefficient of SAW devices [3,31,32]. In addition, a low-cost SAW device based on a thin layer of PZT deposited on a non-piezoelectric substrate was reported [33], but detailed studies on the use of thin films of lead-free ferroelectric materials for gas sensing in SAW devices are scarce. Multi-component systems of lead titanate (PbTiO_3_) and lead zirconate (PbZrO_3_), called lead zirconium titanate (Pb[Zr_x_Ti_1−x_]O_3_ (0 ≤ x ≤ 1)—PZT), are the most widely employed perovskite ceramic materials due to their advantageous properties, such as greater sensitivity and higher operating temperature, compared to other piezo ceramics [34,35]. Since 2006, legislation restricts the use of materials that pose risks to human health and to the environment, thus lead-based materials are considered to be particularly toxic and, in this regard, the attention of researchers has turned to other lead-free piezoelectric materials.

Among this class of eco-friendly materials, barium titanate (BaTiO_3_-BT) was the first ceramic material with ferroelectric properties ever discovered, but the interest in it declined after the discovery of PZT, whose piezoelectric properties were superior [36]. The potential of using BT-based ceramics in piezoelectric applications was reconsidered after 2009, when Liu and Ren found a surprisingly high piezoelectric coefficient (d33∼620  pC/N) for the non-lead ceramic system Ba(Ti_0.8_Zr_0.2_)O_3_−(Ba_0.7_Ca_0.3_)TiO_3_ [37]. This strategy for improving the piezoelectric response of BT by introducing Ca^2+^ and Zr^4+^ into its crystal structure has led to increased research papers on this topic [38]. Lead-free piezoelectric materials (Ba, Ca)(Ti, Zr)O_3_ (BCTZ) with different properties were obtained by varying the concentrations of the two ferroelectric oxides Ba(Ti_0.8_Zr_0.2_)O_3_ and (Ba_0.7_Ca_0.3_)TiO_3_, which enables their use in applications such as energy harvesting and storage or actuators. Different techniques for growing thin layers of BCTZ on various substrates have been reported in the literature, from chemical ones, like the sol-gel method [39], to physical ones, namely, RF magnetron sputtering [40] and pulsed laser deposition [41].

Using polymer-inorganic composites was demonstrated to result in high-performance gas sensors due to the synergistic effects [42] between specific functional materials when used with an adequate fabrication technology. Numerous polymers can be used as sensing materials, for instance, polypyrrole (PPy), polyaniline (Pani) or polythiophene and their derivatives [43,44]. In this present work, we selected polyethylenimine (PEI) to obtain hybrid layers of polymer-inorganic composites.

Motivated by the possibility of using lead-free piezoelectric/ferroelectric materials for SAW devices [3], [33], here we report the fabrication and functional properties of surface acoustic wave (SAW) sensors based on thin films of (1 − x) Ba(Ti_0.8_Zr_0.2_)O_3−x_(Ba_0.7_Ca_0.3_)TiO_3_ (BCTZ50) in combination with a Polyethylenimine (PEI) layer, using laser-based deposition techniques to obtain both inorganic and organic layers. As the various characteristics of an active sensor element (i.e., thickness, crystalline structure, surface roughness, chemistry, etc) can influence the response of an SAW sensor to a specific analyte, the thin layers of BCTZ50 were deposited by pulsed laser deposition (PLD) onto different monocrystalline substrates (SrTiO_3_, GdScO_3_) to induce different epitaxial characteristics (strain, texture) into the film’s crystalline structure. This was followed by depositing a PEI polymer on top of the epiatxial BCTZ50 thin films using the matrix assisted pulsed laser evaporation (MAPLE) technique. The structures obtained were further integrated into SAW devices to evaluate their sensing properties in different gaseous environment.

## 2. Materials and Methods

### 2.1. Deposition of Thin Films 

The (1 − x) Ba(Ti_0.8_Zr_0.2_)O_3−x_(Ba_0.7_Ca_0.3_)TiO_3_ (BCTZx) thin films were deposited by pulsed laser deposition (PLD) using the x = 50 composition.

BCTZ50 targets were prepared via the conventional sintering route, starting from precursors and followed by milling, pressing and sintering at high temperature (1450 °C).

The substrates used for deposition were monocrystalline Strontium Titanate (SrTiO_3_ or STO) and Gadolinium Scandium Oxide (GdScO_3_ or GSO). For the fabrication of the BCTZ50/STO and BCTZ50/GSO samples, an ArF excimer laser was used, the laser fluence value being 1.65 J/cm^2^. The number of laser pulses ranged between 8000 and 36,000, with a 5 Hz repetition rate. The substrate temperature was set to 700 °C, with a heating rate of 50 °C/min while the cooling rate was set to 10 °C/min. The flow of oxygen during the ablation process was set to 20 sccm and the gas pressure reached 1 × 10^−1^ mbar, before starting laser ablation. When the ablation process ended, the cooling of the substrates was performed in oxygen flow, set to 200 sccm.

The Polyethylenimine (PEI) polymer was deposited using matrix assisted pulsed laser evaporation (MAPLE). The MAPLE deposition method involves obtaining a target from a polymer dissolved in a solvent (matrix). The PEI target was obtained by dissolving 400 mg of PEI in a 1:1 solution of deionised water and isopropyl alcohol, under magnetic stirring, for 60 min. The target thus obtained was frozen in liquid nitrogen and further irradiated, using 266 nm of a Nd:YAG laser, at 10 Hz with 36,000 laser pulses.

The surface acoustic wave sensors (SAW) were used to evaluate the response of BCTZ thin film covered and uncovered with PEI. For the gas response measurement, interdigital electrodes were deposited on the surface of thin films via thermal evaporation.

For the IDT, Au metal electrodes were deposited, with thicknesses of approximately 200 nm on the BCTZ thin layers. Later, gold wires with a diameter of 75 µm were glued to the electrodes, using an epoxy conductive silver (Ag). The sensors were tested and characterised for different types of usual gases (O_2_, CO_2_ and N_2_), at room temperature and with a humidity of 60%. The humidity was monitored using a hygrometer.

Using the methods and techniques described above, an evaluation of the gas response of BCTZ50 and PEI/BCTZ50 structure was conducted.

### 2.2. Thin Layer Characterization

In this study, we explored the surface morphology of the samples through scanning electron microscopy (SEM). Our analyses were performed using the Scios 2 DualBeam, an advanced ultra-high-resolution analytically focused ion beam scanning electron microscopy FIB-SEM system (Thermo Fisher Scientific Inc. in Hillsboro, OR, USA), at voltages reaching up to 30 kV. The topography of the probes was further analysed by employing atomic force microscopy (AFM) (XE 100 AFM, Park systems KANC 15F, Gwanggyo-ro, Suwon, Republic of Korea) and the measurements were performed in non-contact mode.

Optical properties were investigated by spectroscopic ellipsometry measurements on a Woollam Variable Angle Spectroscopic Ellipsometer (VASE) system equipped with a high-pressure Xe discharge lamp. The spectral range analysed was 1–5 eV, representing the near IR to the UV (260–1200 nm) spectrum and was carried out at a fixed angle of incidence. For obtaining the refractive indexes and extinction coefficients of the analysed layers, WVASE32 software (VASE, J.A. Woollam Co., Inc., Lincoln, NE, USA) was used for fitting and extracting the data from complex multilayer response.

High-resolution X-ray diffraction was conducted on a PANalytical X’pert MRD system (Almelo, The Netherlands) using a parallel monochromatic beam of CuK_α1_ (λ = 1.540598 Å) provided by a four-bonce crystal of Ge(220) placed in the incident beam.

### 2.3. SAW Measurements

The SAW testing system was composed of an amplifier (DHPVA-200 FEMTO amplifier—Messtechnik GmbH, Berlin, Germany), a frequency counter (CNT-91 Pendulum—Spectracom Corp, Rochester, NY, USA) connected to a computer with specialised software. The gas flow was ensured by a mass flow controller, through a mass flow meter, and the gas concentration used for testing was: N_2_—99.996%; CO_2_—99.998%; and O_2_—99.999%; with a gas flow of 0.5 L/min. All investigations were performed at room temperature.

## 3. Results and Discussion

### 3.1. AFM and SEM Morphological Characterization

Scanning electron microscopy and atomic force microscopy were used first to analyse the morphology and roughness of the deposited coatings surfaces. Examples of the SEM images with a magnification of 20 k of the samples obtained by PLD are presented in Figure 1.

The SEM analysis provided valuable insights into the characteristics of thin films deposited using Pulsed Laser Deposition (PLD) on different substrates. Specifically, when applied to Strontium Titanate (STO) substrates, PLD consistently yielded thin films with a uniform distribution across the surface. However, a notable observation was the regular accumulation of material in the form of grains on the surface; a phenomenon commonly associated with PLD technique.

Despite the presence of these surface grains, an interesting finding emerged: the overall appearance and characteristics of the film surfaces did not undergo significant changes even when the number of pulses was varied. This suggests a robustness in the PLD-deposited films on STO substrates, as their fundamental properties appeared to remain stable under different deposition conditions.

In contrast to the STO substrates, the examination of Barium Calcium Titanate Zirconate (BCTZ) films on Gadolinium Scandium Oxide (GSO) revealed a distinctly different outcome. The SEM analysis highlighted the remarkably smooth surface morphology for the BCTZ film on GSO. This indicates that the interaction between the PLD technique and the GSO substrate resulted in a unique deposition behaviour, leading to a more even and homogeneous film surface.

These observations not only contribute to our understanding of the PLD process, but also underscore the significance of substrate selection in determining the final morphology of thin films. The ability to control and manipulate the surface characteristics of deposited films is crucial for various applications.

For a better visualization of the surface topography, and to understand the material organization on the surfaces, AFM measurements were performed.

The influence of the substrate can easily be seen through the appearance of columnar grains of maximum 40 nm, in the case of the thin film deposited on GdScO_3_, compared with the one on SrTiO_3_, where the surface morphology became smoother. In all cases, the surface was uniformly covered by the deposited thin films, with no defects, and the roughness of the layers was found to be between 4 and 40 nm.

In contrast with the BCTZ samples, the main characteristic for the PEI/BCTZ layered coatings, as confirmed by both SEM and AFM analysis, is a low roughness on the surface, as can be seen in Figure 1b and Figure 2c. MAPLE-deposited PEI layers were smooth, with no cracks, and the two employed analysis techniques revealed uniformly distributed films, covering the entire surface of the probes, in accordance with other studies published in the literature [45].

### 3.2. XRD Measurements

The films were deposited on (001)-oriented cubic STO substrates and (110)-oriented orthorhombic GSO substrates, respectively. For simplicity, and for comparison, we considered a pseudocubic lattice for GSO. The orthorhombic (110) orientation of GSO is equivalent to the (001) of a pseudocubic symmetry, and consequently, the substrates were prescribed the subscript “pc” for GSO and “c” for STO. STO has a cubic lattice parameter of 3.905 Å, and GSO has a pseudocubic lattice parameter of 3.967 Å [46]. Figure 3 displays the conventional X-ray 2*θ-θ* scans for the BCZT films on *(b)* STO and *(c)* GSO substrates. The x = 50 value for BCZT target is, at room temperature, near the morphotropic phase boundary (MPB) separating the rhombohedral (R) and tetragonal (T) phases by an intermediate orthorhorombic (O) phase. The target BCZT50 XRD patterns exhibits a distorted orthorhombic phase which we had refined in a cubic symmetry (S.G. Pm-3m) to obtain a pseudocubic lattice parameter of 4.0118 Å [47], consistent with the standard BCZT powder XRD pattern, ICDD card no. 00-063-0614 (Figure 3a). All the films show only the (00*l*) reflections with no secondary phase, revealing a fully epitaxial growth. The epitaxial coherent growth of the BCZT films is confirmed by the four-circle Φ scans around (101) BCZT and their corresponding (101)_c_ STO and (101)_pc_-GSO reflections, respectively. For all the films, a fourfold symmetry is clearly seen (Figure 4).

The out-of-plane and in-plane lattice parameters were determined from conventional and off-axis scans. The mosaicity of the films is described by the ω-scans of the (00*l*) symmetric reflections. Figure 5 presents the superimposed ω-scans of the (002) reflections of the BCZT50 films deposited on STO and GSO. From the evolution of the broadening of these ω-scans of the (00*l*) reflections, using the Williamson–Hall approach proposed by Mentzger et al. [48], the lateral coherence length, which is parallel to the substrate (L_║_), and the mean mosaic tilt angle (α_tilt_) were extracted. We employed the approach described in our previous works [41,47]. The structural data are gathered in Table 1, along with the values of the degree of the in-plane strain due to the misfit relative to the substrate, *ε_in-plane_*, calculated as (*a_in-plane_* − *a*_substrate_)/*a*_substrate_ ∗ 100%. 

An examination of the structural data collected in Table 1 shows the results to be related to the nature of the substrate (STO or GSO) and to the thickness of the films controlled by the used number of laser pulses. Furthermore, one can see that the smallest in-plane strain values are obtained for the thin films deposited onto GSO, due to GSO’s larger pseudocubic lattice parameter. Apparently, the thickness of the thin films affects their mosaicity: the lateral coherence length increases with the thickness while the mean mosaic tilt angle decreases.

With the improved mosaicity, the crystallinity of the thicker-strained relaxed films is clearly emphasised by the broadness of rocking curves in the BCZT (002) plane, as presented in Figure 5. However, for the thin films deposited onto GSO substrates, one should bear in mind the actual orthorhombic symmetry of the substrate which induces anisotropic strains in the BCZT films, which generates interesting structural features [49].

### 3.3. Optical Properties: Spectrometric Ellipsometry (SE)

Ellipsometry measurements were carried out to evaluate the optical properties of BCTZ samples by performing measurements in the 260–1200 wavelength nm range, at room temperature. The Δ and *ψ* parameters are the phase difference and amplitude ratio in the electric field; they describe the change in polarization (from circular to elliptic polarization) and were fitted by building an optical model. The change in polarization state occurs when the incident light beam interacts with measured structure. Δ and *ψ* depend on the optical properties of the analysed structures and the thicknesses of material layers. In this case, the samples were thin layers of BCTZ deposited on monocrystalline substrates (SrTiO_3_ and GdScO_3_). The optical model consisted of a stack of three layers: the substrate, the deposited BCTZ layer and the rough top layer. Each layer was characterised by its own dielectric function (optical properties). For the monocrystalline substrates, we performed the SE measurement before the PLD deposition and we calculated the dispersion of “*n*” and “*k*” by fitting Δ and *ψ* and using point-by-point regression analysis [50]. The BCTZ layer was fitted by a single Gauss oscillator and the top rough layer was assumed to consist of 50% air and 50% BCTZ in a Bruggeman approximation.

The calculated thicknesses of the BCTZ layer and the value of roughness are presented in Table 2. In the case of BCTZ50 deposited with 8000 laser pulses, thicknesses values between 60 and 100 nm were obtained, with a roughness of a few nanometres. When the number of laser pulses increased to 36,000, the obtained thickness was more than three times higher. The was no linear relation between laser pulses and the calculated thickness, even if the rest of deposition parameters were kept constant (oxygen pressure, distance between target and substrate, laser fluency and substrate temperature) during the PLD process.

When using 36,000 laser pulses, the thickness increased a few times and there were no significant differences between the values of refractive indexes in the visible-IR spectral zone (Figure 6). Nevertheless, the extinction coefficient values were slightly different. For the 8000 BCTZ sample, “*k*” was higher (*k*~0.6 at λ = 300 nm, compared to *k*~0.4 for 36,000 pulses) and the difference can be explained by the strain induced in the thin film caused by the difference between the SrTiO_3_ substrate lattice constant (cubic with a = 3.905 Å) and the lattice constant of BCTZ50 (*pseudo-cubic with a(pc) = 4.0118 Å*). When the sample thickness increased, the thin film relaxed, and the properties were found to be similar with bulk BCTZ. For BTCZ thin film deposited on STO, because there is a difference between the Zr and Ca atom radii, which led to slightly different packing density, we obtained different values of optical constants.

In the case of a BCTZ thin layer grown on GdScO_3_ monocrystalline substrate, a similar optical behaviour to the BCTZx grown on STO was obtained (Figure 7).

For the band gap calculation, Tauc plotting was employed [46]. By plotting the absorption coefficient α (α = 4πk/λ) as a function of photon energy (eV), the values of the band gap are obtained. BaTiO_3_ (BTO) in a tetragonal structure presents with an indirect band gap [47], with an energy value of 3.41 eV. Instead of BTO, the value of the indirect band gap for BCTZ50 grown on STO substrate was found be higher, E_gindirect_ = 3.54 eV.

In the case of a BCTZ thin layer grown on a GdScO_3_ monocrystalline substrate, a similar optical behaviour to the BCTZx grown on STO was obtained. The value of the refractive index at λ = 600 nm for BCTZ50 was *n* = 2.311, instead of *n* = 2.332 for BCTZ50/STO. The values of the GSO extinction coefficients were found to be higher for samples grown on STO: a value of *k* = 0.45 for BCTZ50/GSO and *k* = 0.7 for BCTZ50/STO, respectively. The calculated indirect band gap was found to be Eg = 3.58 eV for BCTZ50/GSO. The discrepancy between the optical constants values for samples grown on different substrates is explained by the crystallinity features of the thin films, induced by the substrate during PLD deposition process. For GdScO_3_, the lattice parameters are *a* = 5.52 Å, *b* = 5.79 Å and *c* = 8.03 Å (orthorhombic system) and the accommodation of BCTZ structure on the GSO substrate is different, as compared with cubic STO, leading to different optical properties.

The optical packing density of the BCTZ samples were calculated from experimental values of “*n*” for BCTZ thin films (calculated from SE) and bulk values for the refractive index (*n* = 2.42) for pure BaTiO_3_ [51] by using [52]:(1)$$P=\left(\frac{\left(n_{f}^{2}-1\right)}{\left(n_{f}^{2}+1\right)}\right)\times \left(\frac{n_{b}^{2}+2}{n_{b}^{2}-1}\right)$$
and the porosity ratio:(2)$$P_{or}=1-\left(\frac{n_{f}^{2}-1}{n_{b}^{2}-1}\right)$$

The values are presented in Table 3.

The BCTZ50/STO samples exhibit the highest values for packing density and lower porosity values, in contrast with the BCTZ50 samples deposited on GSO substrate, for which a lower value for packing density (0.956) and a high porosity of 10.62% were calculated.

### 3.4. SAW Measurements

The SAW sensor testing system consisted of an amplifier and a frequency counter connected to a computer with specialised software (Figure 8). The gas flow was ensured by a mass flow controller through a mass flow meter.

The gases used for the tests were carbon dioxide (CO_2_), nitrogen (N_2_) and oxygen (O_2_), and the tests were carried out at a concentration of 99.996% N_2_, 99.999% O_2_, 99.998% CO_2_ and a gas flow of 500 sccm.

The first important aspect noted from the frequency shifts was that the sensors with a PEI-sensitive layer had a higher frequency shift than those without a polymeric layer. At the same time, the sensors based on BCTZ50 deposited on an SrTiO_3_ substrate had a small response or did not exhibit any response without the polymeric sensitive layer. A higher response was obtained for BCTZ50 deposited on a GdScO_3_ substrate with a frequency shift between 12–19 KHz (Figure 9). In the case of BCTZ50/SrTiO_3_, when the thickness was increased by a higher number of laser pulses, the response of the SAW sensor became unclear and very noise and we cannot accurately state the existence of a gas response (for example BCTZ50 deposited with 36,000 laser pulses in Figure 10). The test response from the PEI/BCTZ50/STO (8000 pulses), presented in Figure 11, sensor indicates a greater efficiency when testing for N_2_, with 12.66 kHz frequency shift, compared to responses of 9 kHz for O_2_ and 11.29 kHz for CO_2_.

The best results for the frequency shifts were obtained in the case of PEI/BCTZ50/GSO thin films. The highest frequency shift was about 39 kHz, obtained for O_2_ and CO_2_ gases by the PEI/BCTZ50/GSO-based SAW sensor. The values of the frequency shifts for these sensors were not contrasting, even when comparing different types of gases tested, which means that we cannot state if they are selective sensors.

The response time of the sensors was also influenced by the presence of the polymeric layer. Although we observed that, for BCTZ50/GSO sensors, the frequency shift was improved with the use of the polymeric coating, the influence on the response time was both increasing and decreasing. This increase in response time is explained by the fact that the adsorption of molecules at the level of the sensitive layer takes place more slowly than when gas molecules come into direct contact with the sensor substrate. However, considering that the frequency shift increases when using the polymer, an increase in the response time is allowed. The BCTZ50/GSO sensor, in the tests for CO_2_, obtained both a decrease in the response time and an increase in the frequency shift (Figure 9).

The mechanism of gas sensing was explained by Wang et al. [53]. When gas molecules interact with surface of an oxide, the charge state is altered, and the conductivity of sample is changed. A change in the conductivity of layer leads to a change in SAW sensor response. For all samples deposited with 8000 laser pulses, we calculated, from optical analysis, a percentage of pores around 6–8%. From Figure 10 and Figure 11, we can observe the response for BCTZ50/STO without a polymeric layer in the presence of two analysed gases (O_2_ and CO_2_). The presence of pores causes oxygen to be easily adsorbed and the high porosity provides a larger amount of surface sites for gas adsorption and chemical reactions [54]. This may attract more electrons from the BCTZ layer and substantially modify their frequency response in SAW measurement configuration. This behaviour in BaTiO_3_-based materials was already reported by Park et al. [55] in the case of Sb_2_O_3_-BaTiO_3_, and they showed the importance of porosity for gas response in the case of Sb2O_3_-BaTiO_3_. Moreover, other studies reported similar response behaviours for CO sensing, such as in the case of SnO_2_, which has higher sensitivities but a smaller grain size [56,57,58].

By ading a polymer layer (PEI in this case), the mecanism behind overall response was changed. This time, the gas molecules were absorbed by the PEI layer, and because BCTZ is a piezoelectric material, changing the mass of the polymeric cover layer induced a response in the BCTZ film. The frequency shifts in samples with a polimeric layer were around 9–11 KHz for CO_2_ and O_2_, and this is normal behavior for PEI [59].

When the number of laser pulses increased to 36,000 and the thickness of BCTZ50/STO layer increased more than three times, a strange behaviour was observed in the gas response. For N_2_ in the case of the BCTZ50/STO with a thickness of ~300 nm the frequency response was noisy, but when N_2_ was introduced in testing chamber, a shift was observed (Figure 10). For CO_2_ and O_2_ the shift was noticeable but weak and the frequency response became noisy and unstable in a matter of a few seconds after the gas was introduced. Even with PEI layer, the gas response was not clear. At high thicknesses, the properties of the BCTZ thin layer were similar to bulk BCTZ, and because there was no gas response for thicker BCTZ50/STO, we can conclude that this material is only suitable for gas detection when the functional properties are altered by structural strain.

In the case of BCTZ50 thin films deposited on GSO, the gas response as a function of thickness/strain relaxation was completely different in respect to the BCTZ50/STO films. For the thin, 8000 laser-pulsed BCTZ50/GSO sample without the PEI coating, only a weak response in the presence of CO_2_ was noticed. For the BCTZ 50/GSO thin films, the frequency measured during time before and after gas was introduced to the chamber are presented in Figure 12 and Figure 13. For the thickest BCTZ50/GSO sample, a good response was noted without the polymer layer for all analysed gases, with a frequency shift between 12 and 39 KHz. The best response time for the thick BCTZ50/GSO sample was around 80 s in the case of oxygen gas detection. By adding a polymer layer, the frequency shift was increased to ~50 KHz; a higher frequency shift translated into a better response for all gases, but for CO_2_, the response time was also increased a few times. The highest response time can be explained by a slow chemical reaction between the polymer and the gas molecules.

The difference between the gas responses from BCTZ deposited on different substrates were clearly observable, and in close correlation with crystalline features of the samples. As the in-plane strain-ɛ (%) in the BCTZ50/GSO samples is much smaller than in the case of the BCTZ50/STO samples, combined with the improved moisaicity (FWHM-ω(002)), it is clear that the less-structurally defective BCTZ50/GSO films are suitable for acoustic wave propagation within the films in contrast with the BCTZ50/STO ones.

## 4. Conclusions

Here, gas sensing performance as a function of thickness and substrate variation in BCTZ50 and PEI/BCTZ50 thin films deposited by laser-based techniques, in correlation with morphological, structural and optical characteristics, has been presented. BCTZ50 samples with different thicknesses were deposited on two types of substrates with different crystal structures, cubic (SrTiO_3_) and orthorhombic (GdScO_3_). Spectrometric ellipsometry investigations revealed that, when the sample thickness increased, the thin film relaxed, and the optical properties were found to be similar with bulk BCTZ. The values of the GSO extinction coefficients were found to be higher for samples grown on STO, and the difference between optical constants values for samples grown on different substrates can be explained by the crystallinity of thin films, induced by the substrate during the PLD deposition process. After depositing IDT Au electrodes, the thin films were integrated into SAW devices in order to measure the frequency response to various gases (N_2_, CO_2_ and O_2_). The response of BCTZ50 and PEI/BCTZ50 sensors was evaluated. The first important aspect noted from the frequency shifts was that the sensors with a PEI-sensitive layer had a higher frequency shift than those without a polymeric layer. At the same time, the sensors based on BCTZ50 and deposited with 8000 laser pulses on a SrTiO_3_ substrate did not have such a high response without a polymeric sensitive layer. The test response of the BCTZ50/GSO sensor was 17.6 kHz with PEI and 19.1 kHz without PEI, indicating a greater efficiency when testing for N_2_. The best efficiency when testing for O_2_ was recorded for PEI/BCTZ50, with a value of 39.5 kHz, more than two times higher compared to the film without the polymeric layer. The best results of the frequency shifts were obtained by the PEI/BCTZ50 sensors deposited on GdScO_3_; this was also the best when detecting CO_2_, with a recorded frequency shift of 39 kHz. In the case of CO_2_ detection, when using PEI, the frequency shift increase was more than three times recorded for bare BCTZ50/GSO. The response time of the sensors was also influenced by the presence or absence of the polymeric layer, noting an increase in the response time when using PEI. This behaviour resulted from the fact that the adsorption of molecules at the level of the sensitive layer takes place more slowly than when gas molecules come into direct contact with the sensor substrate, thus an increase in the response time with an increase in the frequency shift is acceptable.

## Figures and Tables

**Figure 1 nanomaterials-14-00039-f001:**
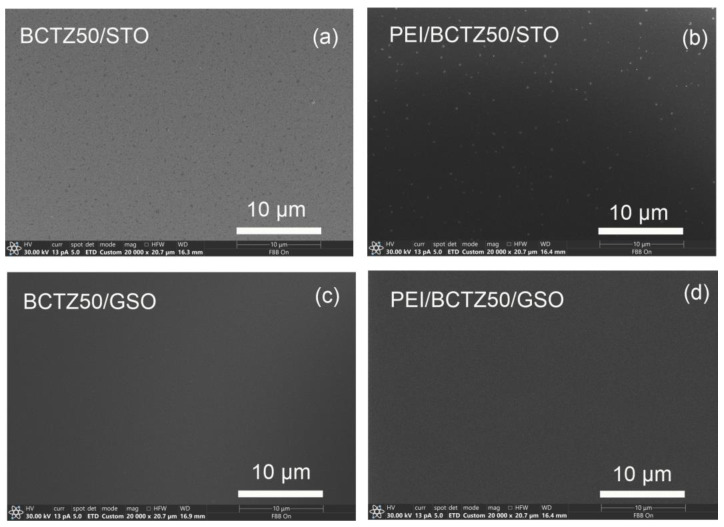
SEM images of thin films deposited on (STO) heated at 700 °C in mbar of oxygen (**a**) BCTZ50/STO; (**b**) PEI/BCTZ50/STO; (**c**) BCTZ50/GSO and (**d**) PEI/BCTZ50/GSO.

**Figure 2 nanomaterials-14-00039-f002:**
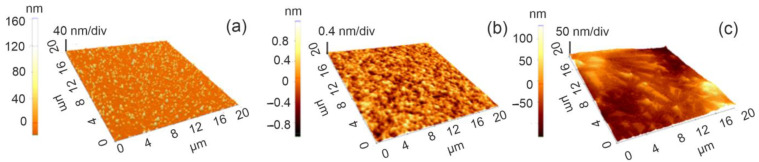
AFM topography images for 36,000 laser pulses BCTZ thin films (**a**) BCTZ50/GSO (**b**) BCTZ50/STO and (**c**) PEI/BCTZ50/STO.

**Figure 3 nanomaterials-14-00039-f003:**
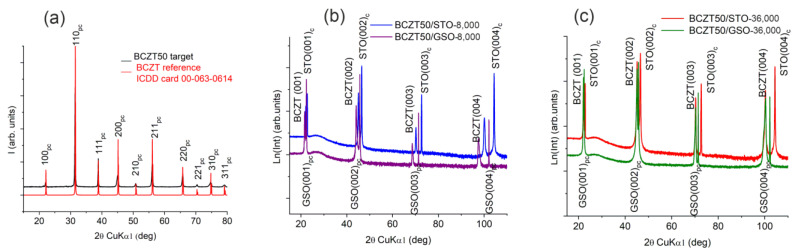
XRD patterns of the BCZT. (**a**) BCTZ standard card ICDID card no. 00-063-0614 and BCTZ50 target; (**b**) thin films deposited on GSO with 8000 laser pulses; (**c**) thin films deposited on STO with 36,000 laser pulses.

**Figure 4 nanomaterials-14-00039-f004:**
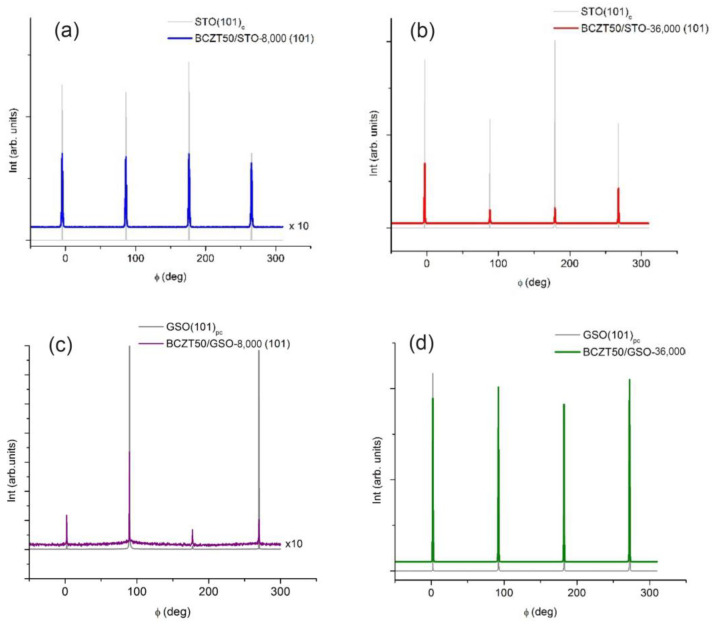
XRD four-circle Φ scans around (101) BCZT reflection for (**a**,**b**) BCTZ/STO films with different thickness and (**c**,**d**) BCTZ/GSO films with different thickness.

**Figure 5 nanomaterials-14-00039-f005:**
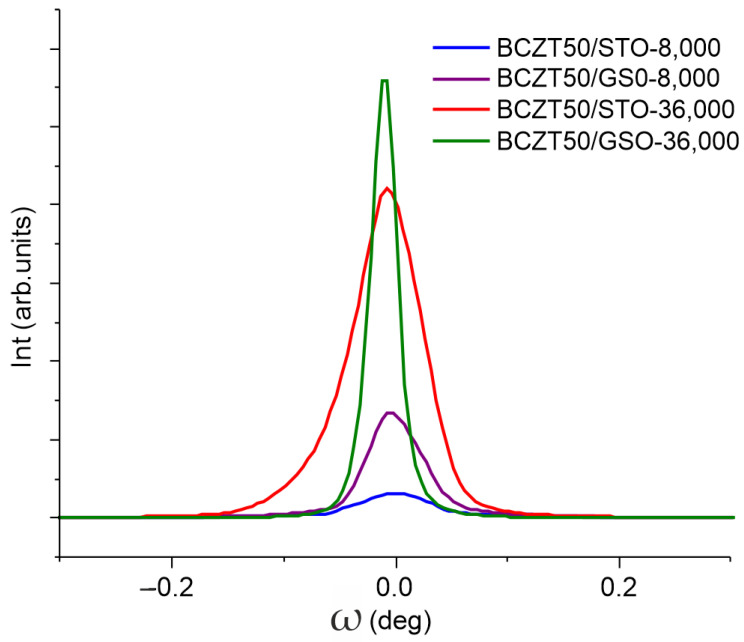
XRD X-ray rocking curves (ω-scans) of the BCZT (002) reflections for the films deposited on STO and GSO substrates.

**Figure 6 nanomaterials-14-00039-f006:**
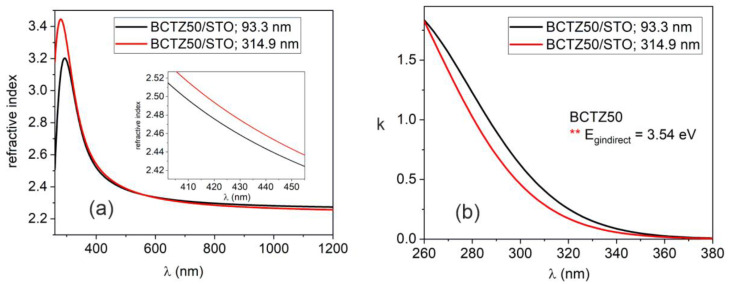
Comparison between the dispersion of optical constants for the BCTZ50/STO thin layer growth at 8000 and 36,000 laser pulses: (**a**) refractive index and (**b**) extinction coefficients. In caption zoom of refractive index for samples in UV spectral region

**Figure 7 nanomaterials-14-00039-f007:**
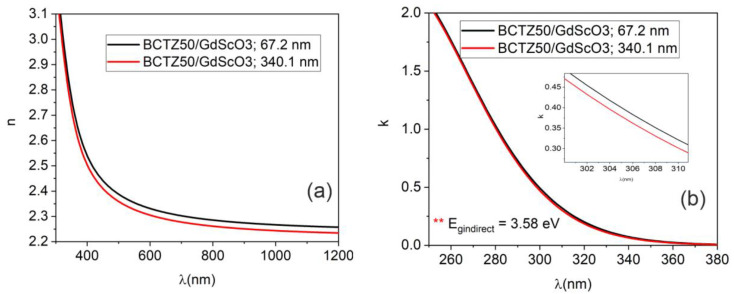
The values of optical constants for the BCTZ50 thin layers deposited on GdScO_3_ substrate at (**a**) 8000 and (**b**) 36,000 laser pulses: (**a**) refractive index and (**b**) extinction coefficients. In caption zoom of exctinction coefficients for samples in UV spectral region

**Figure 8 nanomaterials-14-00039-f008:**
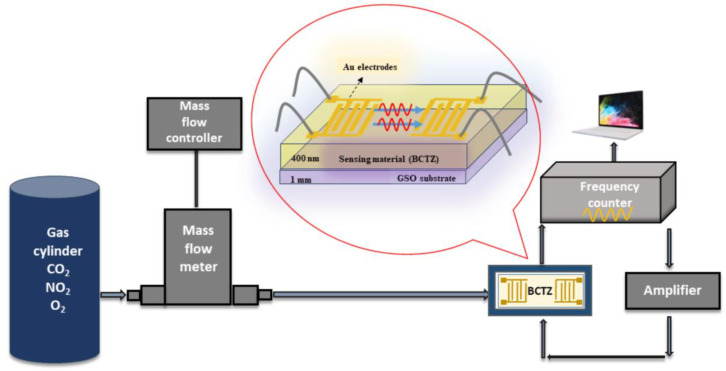
Experimental setup for SAW sensor frequency shift measurements.

**Figure 9 nanomaterials-14-00039-f009:**
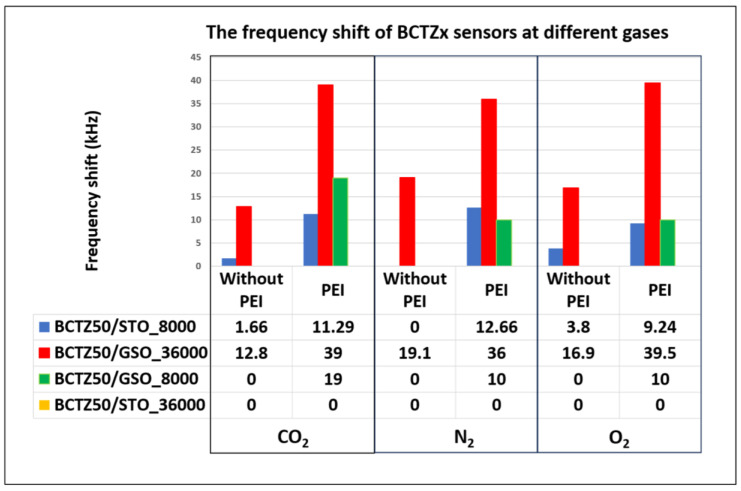
The frequency shift of BCTZ*x* sensors at different gases.

**Figure 10 nanomaterials-14-00039-f010:**
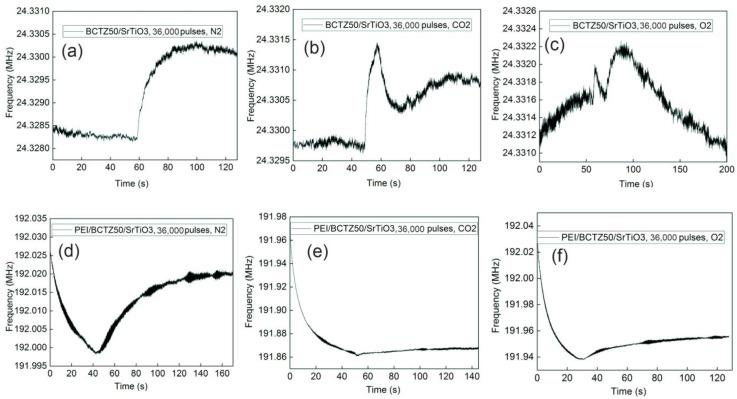
The gas response for the BCTZ50 deposited on SrTiO_3_ substrate by PLD with 36,000 laser pulses without polymers for (**a**) N_2_; (**b**) CO_2_; (**c**) O_2_; and PEI/BCTZ50 for (**d**) N_2_; (**e**) CO_2_; (**f**) O_2_.

**Figure 11 nanomaterials-14-00039-f011:**
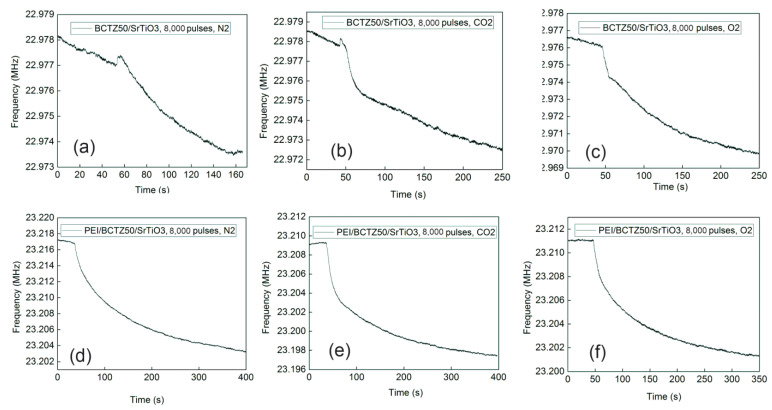
The gas response for the BCTZ50 deposited on SrTiO_3_ substrate by PLD with 8000 laser pulses without polymers for (**a**) N_2_; (**b**) CO_2_; (**c**) O_2_; and BCTZ50 with PEI polymer for (**d**) N_2_; (**e**) CO_2_; (**f**) O_2_.

**Figure 12 nanomaterials-14-00039-f012:**
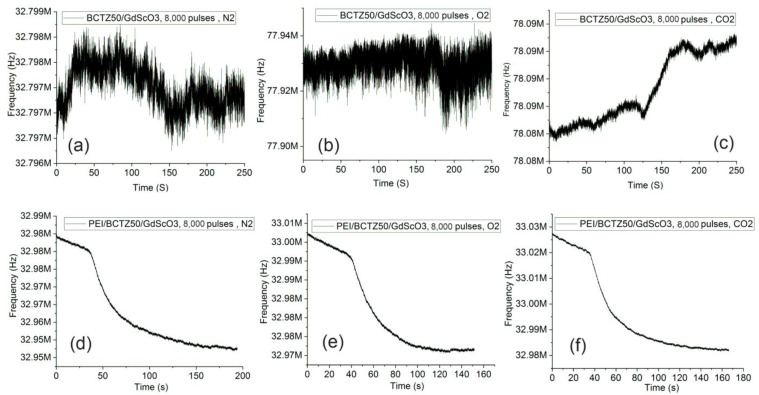
The gas response for the BCTZ50 deposited on GdScO_3_ substrate by PLD with 8000 laser pulses without polymer for (**a**) N_2_; (**b**) CO_2_; (**c**) O_2_; and PEI/BCTZ50 for (**d**) N_2_; (**e**) CO_2_; (**f**) O_2_.

**Figure 13 nanomaterials-14-00039-f013:**
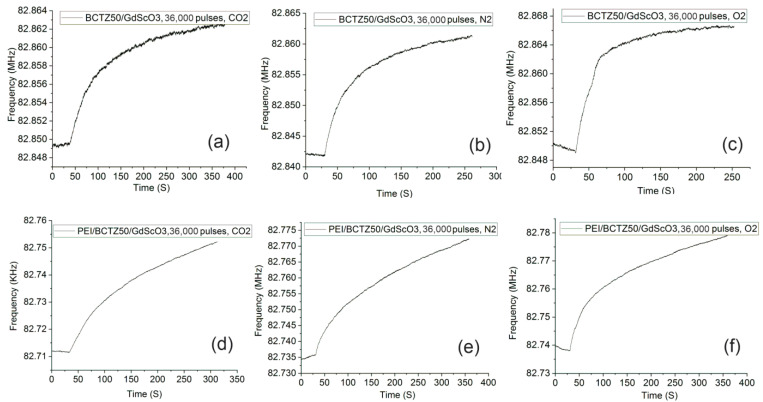
The gas response for the BCTZ50 deposited on GdScO_3_ substrate by PLD with 36,000 laser pulses without polymer for (**a**) N_2_; (**b**) CO_2_; (**c**) O_2_ and PEI/BCTZ50 for (**d**) N_2_; (**e**) CO_2_; (**f**) O_2_.

**Table 1 nanomaterials-14-00039-t001:** Structural data extracted from XRD analysis.

Probe	Laser Pulses	*a_out-of-plane_* (Å)	*a_in-plane_* (Å)	In-Plane Strain-*ε _in-plane_* (%)	FWHM-*ω*(002)(deg)	L_II_ (nm)	α_tilt_ (deg)
BCZT50/STO	8000	4.0193	4.0173	2.88	0.068	72	0.227
BCZT50/GSO	8000	4.1022	3.9799	0.33	0.056	270	0.096
BCZT50/STO	36,000	4.0152	3.9857	2.07	0.078	376	0.051
BCZT50/GSO	36,000	4.0162	4.0147	1.20	0.031	471	0.054

**Table 2 nanomaterials-14-00039-t002:** The values of thicknesses and the Gauss parameters for the BCTZ samples growth by PLD.

Probe	Laser Pulses	Thickness (nm)	Roughness (nm)	Amp Gauss	En (eV)	Br (eV)	MSE
BCTZ 50/STO	8000	93.3	2.1	9.31	4.70	1.02	2.777
BCTZ 50/STO	36,000	314.9	1.3	12.29	4.93	1.12	1.374
BCTZ 50/GSO	8000	67.2	2.8	11.39	4.92	1.17	2.803
BCTZ 50/GSO	36,000	340.1	9.1	11.94	4.89	1.08	8.395

**Table 3 nanomaterials-14-00039-t003:** The packing density and porosity for samples of BCTZx deposited by PLD on monocrystalline substrate.

Probe	Laser Pulses	Refractive Index (λ = 600 nm)	Packing Density	Porosity (%)
BCTZ50/GSO	8000	2.33	0.961	8.8
BCTZ 50/STO	8000	2.335	0.966	8.34
BCTZ 50/STO	36,000	2.332	0.963	8.62
BCTZ 50/GSO	36,000	2.311	0.956	10.62

## Data Availability

The data presented in this study are available on request from the corresponding author.

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
