# Peer review of "Lead-Free Perovskite Thin Films for Gas Sensing through Surface Acoustic Wave Device Detection"

_nanomaterials, 2023, doi:10.3390/nano14010039_

Round 1

Reviewer 1 Report

Comments and Suggestions for Authors

In this manuscript, the authors have attempted to experimentally identify the BCTZ film’s thickness and crystallographic features on the characteristics and performance of the SAW device. In addition, authors further integrated SAW sensors into the testing system to evaluate the response of BCTZ thin films with PEI to N2, CO2, and O2 gases. After addressing the following questions, this paper could be considered for publication in the journal.

1. The ruler size and font size should be consistent in the SEM image of Figure 1. In Table 5, the font format is inconsistent.

2. Figure 3 should include the XRD standard card of BCTZ50 or the XRD spectrum of the pure BCTZ50.

3. Scanning electron microscopy images of BCTZ50/GSO and PEI/BCTZ50/GSO should be supplemented with explanations.

4. In SAW sensor measurements, the investigations were performed at room temperature and humidity of 60%. To ensure the accuracy of the results, humidity interference factors should be considered.

5. In carbon dioxide detection, the PEI/BCTZ50/GSO sensor showed good results. However, similar frequency shifts also occur when the sensor is exposed to other gas environments (O2/N2), which may affect the accuracy of the sensor. How to avoid gas interference?

Author Response

We thank the Reviewer for his/her valuable suggestions and criticism. We have modified the manuscript (in the manuscript modifications are marked by single-line underlining) in agreement with all the requirements.

Reviewer 2 Report

Comments and Suggestions for Authors

In this research, SAW gas sensors based on lead-free perovskite (1−x) Ba(Ti0.8Zr0.2)O3-x(Ba0.7Ca0.3)TiO3 (BCTZ50, x=50) are reported and the effects of their crystallographic and optical properties on gas sensing are analyzed. The experimental data is sufficient and the analysis process is rigorous. I think this article is worth publishing, but a few questions remain as follows:

1. In the first paragraph of introduction section, lines 42-44, the authors' description is inaccurate because not all SAW sensors require a recognition layer in-between, such as SAW strain sensors. Please check.

2. In order to facilitate the reader's understanding, please give the structural diagram of the prepared SAW gas sensor.

3. In Figure 1, the processing of the two SEM images should be consistent.

4. In Section 3.1, Paragraph 2, lines 179-180, the author described “a common feature for depositions produced by PLD technique”, please cite literatures to prove it.

5. In Section 3.3, Paragraph 1, line 206, the authors' description does not correspond to Figure 3. Please check.

6. Figure 6 is not cited in the paper.

7. In Section 3.4, Paragraph 3, lines 276-279, the authors' description is inconsistent with Figure 6 and Figure 7. Please check.

8. In Section 3.5, Paragraph 4, line 347 and line 349, according the authors' description,BCTZ50/GSO” should be replaced by “PEI/BCTZ50/GSO”.

9. How were the thicknesses obtained for the thin films prepared with different laser pulses in the paper? Please explain.

Comments on the Quality of English Language

Suggest well improve the writing.

Author Response

We thank the Reviewer for his/her valuable suggestions and criticism. We have modified the manuscript (in the manuscript modifications are marked by double-line underlining) in agreement with all the requirements.

Reviewer 3 Report

Comments and Suggestions for Authors

This manuscript reports the synthesis and gas sensing characteristics of (1x) Ba(Ti0.8Zr0.2)O3-x(Ba0.7Ca0.3)TiO3 (BCTZ50, x=50) thin films. The synthesis and characterization of sensing film are relatively well-investigated. The followings should be examined for further consideration.

1.     The background and advantages of using SAW gas sensors should be stressed and provided in the manuscript.

2.     ε in Table 1 is marked in red. Please change it to black.

3.     Some words in most Figures are very small.

4.     I recommend that the authors plot the response for each gas using a bar graph. That way, it will be easier to compare the gas response at once.

5.     Page 15, the authors only cited literature to support the frequency response modification in SAW measurement of BaTiO3-based materials. The author needs to strengthen this part further.

Author Response

We thank the Reviewer for his/her valuable suggestions and criticism. We have modified the manuscript (in the manuscript modifications are marked by wave underlining) in agreement with all the requirements.

Round 2

Reviewer 1 Report

Comments and Suggestions for Authors

The author answers well for the initial comments.